# The Association of Asthma with Anxiety, Depression, and Mild Cognitive Impairment among Middle-Aged and Elderly Individuals in Saudi Arabia

**DOI:** 10.3390/bs13100842

**Published:** 2023-10-16

**Authors:** Sameera Abuaish, Huda Eltayeb, Asmatanzeem Bepari, Syed Arif Hussain, Raneem Saad Alqahtani, Waad Saeed Alshahrani, Amjad Hayf Alqahtani, Nada Saad Almegbil, Wafa Nedal Alzahrani

**Affiliations:** 1Department of Basic Sciences, College of Medicine, Princess Nourah Bint Abdulrahman University, Riyadh 11671, Saudi Arabia; hheltayeb@pnu.edu.sa (H.E.); ambepari@pnu.edu.sa (A.B.); 438001484@pnu.edu.sa (R.S.A.); 438001688@pnu.edu.sa (W.S.A.); 438001523@pnu.edu.sa (A.H.A.); 437006791@pnu.edu.sa (N.S.A.); 438001404@pnu.edu.sa (W.N.A.); 2Respiratory Care Department, College of Applied Sciences, Almaarefa University, Riyadh 13713, Saudi Arabia; syhussain@um.edu.sa

**Keywords:** asthma, anxiety, depression, cognitive impairment

## Abstract

Asthma is a common chronic inflammatory condition with increasing global prevalence. There is some evidence highlighting the effect of asthma on brain functioning. In Saudi Arabia, limited studies have examined the relationship between asthma and mental health, including cognition and mood disorders in older adults in particular. In this study, we examine the association between asthma and mental health outcomes in middle-aged and elderly individuals in Riyadh, Saudi Arabia. In a cross-sectional study, 243 subjects were recruited from outpatient clinics between 2020–2021 (non-asthmatic: n = 159, asthmatic: n = 84). The Montreal Cognitive Assessment test, the Hospital Anxiety and Depression Scale, and the Asthma Control Test were used to assess cognition, anxiety and depression, and asthma control, respectively. Logistic regression analysis while controlling for covariates revealed an association between asthma and symptoms of anxiety and depression (OR = 2.40 [95% CI: 1.07–5.35]) and mild cognitive impairment (MCI) (OR = 1.80 [95% CI: 1.00–3.24]). Poorly controlled asthma increased the odds of anxiety cases (OR = 4.88 [95% CI: 1.09–17.2]). Stratifying analysis by age intervals revealed that asthma was associated with symptoms of anxiety and depression (OR = 2.5 [95% CI: 1.00–6.08]) in middle-aged patients only, while elderly asthmatics had increased odds of having MCI (OR = 7.4 [95% CI: 2.34–23.31]). These findings highlight the possible effects of asthma and its control on mental health among middle-aged and elderly individuals in Saudi Arabia.

## 1. Introduction

Asthma is a chronic non-communicable disease characterized by chronic inflammation of the airways, affecting over 200 million people globally, with estimates suggesting an increase in prevalence over time [1,2]. In Saudi Arabia, the prevalence of asthma has increased from 4% to 11% in adults diagnosed with asthma by a doctor [3,4]. According to the global asthma report in 2022, asthma is ranked 34th among the leading causes of burden of disease [1].

Recent preclinical studies have suggested the presence of a lung–brain axis in asthma, indicating that brain health is sensitive to chronic inflammation and reduced oxygen levels due to asthma attacks [5,6]. Brain imaging in asthmatic patients has shown structural and functional changes in the brain, which could explain the reported association between asthma and psychological health [7,8,9,10]. Studies have reported a co-morbidity between asthma and anxiety and depression in adults [11,12]. Persistent asthma has also been associated with anxiety and depression [13]. Anxiety and depression among asthma patients are negatively associated with quality of life and asthma control [14,15,16].

The impact of asthma on cognitive function is not fully understood, but research suggests that asthma is associated with reduced cognitive performance. A meta-analysis has indicated that asthma is linked to various cognitive deficits, with a greater impact observed in individuals with severe disease [17]. Asthma is associated with an increased risk of mild cognitive impairment (MCI) and dementia [9,18,19]. Longer asthma duration and lower lung function have also been associated with a higher risk of cognitive dysfunction [20,21]. Additionally, asthma has been found to be associated with smaller hippocampal volumes, which could contribute to cognitive impairment [22]. Overall, these findings highlight the importance of further research in this area and suggest a potential link between asthma and cognitive impairment in adults.

Mental health is an important aspect of overall health, particularly for the older population, who may be more susceptible to cognitive decline and mood disorders. While there are studies examining asthma’s association with mood disorders and mild cognitive impairment, the majority has generally focused on adults as young as 18 years of age, and there are limited studies on middle-aged and older adults. Therefore, this study aims to investigate the association between asthma and anxiety, depression, and mild cognitive impairments in this age group and Saudi Arabia. Also, we examine the relationship between asthma control and anxiety, depression, and mild cognitive impairments. We hypothesize that asthma would be associated with increased odds of anxiety and depression and MCI, and that middle-aged and elderly individuals would be differentially affected by the disease. We also hypothesize that poor asthma control would be associated with worse mental health outcomes.

## 2. Materials and Methods

### 2.1. Study Design and Setting

This study is a single-centric cross-sectional study with a convenience sampling of asthmatics and non-asthmatics from the general population of the capital city of Saudi Arabia, Riyadh. The study was conducted at King Abdullah bin Abdulaziz University Hospital (KAAUH), a public hospital serving mainly university students and employees and their families in addition to the public. Study participants were recruited from outpatient family medicine and pulmonary clinics between 2020–2021.

### 2.2. Population

An a priori sample size calculation was done using the G*Power sample size calculator with a 5% type one error, 95% confidence level, and 80% study power. The calculated sample size required for this study was 194, with 97 participants in each asthmatic and control group. Assuming a higher incidence of controls, a larger sample size of 245 was considered. Two participants were also excluded from the study because they did not complete all the surveys. Therefore, a total of 243 remained for analysis (non-asthmatic: n = 159, asthmatic: n = 84). To ensure a good representation of the sample, a power analysis was conducted with the final sample size obtained, and it indicated an 85% power, demonstrating the sample’s good representativeness. Inclusion criteria were men and women 40 years of age and older with doctor-diagnosed asthma for asthmatic participants. Patients suffering from a severe chronic illness requiring bed rest, physical disability, and the presence of communication barriers were excluded.

### 2.3. Study Procedure

Study participants were approached before or after their scheduled clinic visit to participate in the study. The subjects were escorted into a quiet room where written consent was taken for participation in the study. Afterward, an interview session of 30 min was conducted with each patient to collect demographic data and medical history and assess cognition, anxiety and depression, and asthma control. Patients were asked about their perception of asthma, whether it was mild, moderate, or severe. Medical records were used to collect other medical history and medication use. All study procedures were approved by the Research Ethics Committee of Princess Nourah bint Abdulrahman University, Riyadh (IRB No. 20-0100).

### 2.4. Study Instruments

#### 2.4.1. Anxiety and Depression Assessment

A validated Arabic version of the Hospital Anxiety and Depression Scale (HADS) was used to assess the participant’s depression and anxiety states in clinical settings [23]. The HADS is composed of two subscales assessed by 14 items, 7 items for anxiety and 7 for depression. Each item has a scale from 0–3 and a total of 4 points. The final score was out of 21 for each subscale. Individuals were categorized as normal, borderline case, or case if they scored between 0–7, 8–10, and 11–21, respectively [24]. The anxiety subscale had a Cronbach’s alpha coefficient of 0.80, and the depression subscale had a 0.76—both indicating good reliability in our sample set.

#### 2.4.2. Cognitive Function Assessment

To assess cognition, the validated Arabic version of the Montreal Cognitive Assessment (MoCA) test was used [25]. This test is a validated and highly sensitive tool for detecting MCI [26]. The scale is composed of several cognitive domains to assess cognitive function including visuospatial/executive, naming, memory, attention, language, abstraction, delayed recall, and orientation of time and place. A total cutoff score of <26 is considered mild cognitive impairment.

#### 2.4.3. Asthma Control Assessment

Asthma control was assessed using a validated Arabic version of the asthma control test (ACT), a 5-item questionnaire to assess symptoms, use of medications, and impact on activities in asthmatic patients [27]. Each item is graded on a 5-point Likert-type rating scale, yielding a score from 5 to 25, with higher scores indicating better control [28]. Specifically, scores ≥ 20 indicated well-controlled asthma, while scores in a range of 19–16 represented partially controlled asthma, and scores > 16 indicated poorly controlled asthma [29]. Cronbach’s alpha coefficient was 0.85 for the items of the questionnaire.

### 2.5. Statistical Analysis

Data were analyzed using SPSS (IBM). Descriptive statistics was done for continuous variables and presented as means and ranges, while categorical variables were presented as frequencies and percentages. Data distribution did not show a substantial departure from normality when examining histograms and Q–Q plots. In addition, skewness and kurtosis values were all below one, indicating no extreme departure from normality [30]. Moreover, with sample sizes > 30, studies suggest that parametric tests would not be greatly impacted with moderate deviation from normality [31]. To assess the effects of asthma on anxiety, depression, and MCI, scores of the HADS and MoCA test were analyzed using a univariate ANOVA while controlling for age, gender, and BMI. Participants were later categorized based on the cutoffs mentioned above for each questionnaire. A multinomial logistic regression was carried out to assess the association between asthma and anxiety and depression categories, with the normal category as reference. Another model tested the association between asthma and the presence of any symptoms of anxiety or depression by combining both categories, borderline and cases, into one category, which considers any deviation from normal scores and increases power by combining the two abnormal groups. In addition, we tested the association between asthma and having either anxiety or depression or both using multinomial regression to address the common comorbidity of both psychological disorders. In asthmatic individuals, similar models were carried out to test the association between asthma control and anxiety and depression. A total of four participants did not complete the MoCA test, and they were excluded from the data analysis for the MoCA test. Binomial logistic regression was carried out to assess the association between asthma and MCI. To examine the potential relationship between age and BMI with anxiety, depression, and MCI, a preliminary correlation analysis was conducted, revealing significant correlations. Therefore, age and BMI were included as covariates in all regression models. Additionally, there exists extensive literature demonstrating a link between gender and psychological well-being—with women being disproportionately affected by anxiety, depression, and cognitive impairment [32,33]. As a result of this evidence, gender was also included as a covariate in all regression models. Age was categorized into middle-aged (40–59 years) and elderly (≥60) and used to create four new groups: middle-aged non-asthmatics, middle-aged asthmatics, elderly non-asthmatics, and elderly asthmatics. These groups were used in regression models to assess the association of age intervals and asthma and mood and cognition while controlling for BMI and gender. Odds ratios and 95% confidence intervals were reported. Relationships were considered statistically significant at *p* ≤ 0.05.

## 3. Results

### 3.1. Descriptive Statistics

Table 1 presents the characteristics of the study participants. A total of 243 subjects were recruited in this study, 159 of which were non-asthmatics and 84 who were asthmatics. The majority of participants were females (77%), married (87%), Saudi (91%), employed (60%), and highly educated (61%), with university undergraduate (38%) or postgraduate degrees (23%). The mean age of the study participants was 52 years old with a range of 40–81 years of age. Age intervals were calculated, and participants were categorized as middle-aged (76%) or elderly (24%). Asthma onset in asthmatic individuals was mainly in adulthood after the age of 18 (76%). The prevalence of other chronic conditions including heart disease (7%), hypertension (37%), and diabetes (32%) are presented in Table 1.

### 3.2. Asthma Association with Anxiety and Depression

We first assessed the differences in the mean scores of anxiety and depression between asthmatics and non-asthmatics and found that they were significantly different among the two groups (F_(1,226)_ = 4.28, *p* = 0.04), with asthmatics scoring higher than non-asthmatics (Table 2). On the other hand, depression scores had a non-significant trend indicating some differences between the two groups (F_(1,226)_ = 3.13, *p* = 0.08), where asthmatics reported higher scores compared to non-asthmatics (Table 2). Differences in anxiety and depression scores were significant between asthmatics and non-asthmatics when not controlling for covariates (Appendix A)

Using cutoff scores, all study participants were categorized as either normal (73%), borderline cases (14%), or cases (13%) of anxiety. We examined the prevalence of anxiety amongst asthmatics and non-asthmatics and found that about 26% of asthmatics were within the borderline (16%) and cases (20%) of anxiety compared to 22% of non-asthmatics who were within the borderline (13%) and cases (9%) of anxiety (Appendix A). For depression, about 78% of the study participants were normal, and 16% and 6% were borderline and cases of depression, respectively. The prevalence of depression among asthmatics was 32%, with 24% within the borderline category and 8% within the cases category. On the other hand, about 17% of non-asthmatics scored within the depression cutoff scores, where about 13% scored within the borderline category and 8% within the cases category (Appendix A). About 17% of non-asthmatics exhibited symptoms of anxiety or depression, while 10% exhibited symptoms of both anxiety and depression. In asthmatics on the other hand, 20% and 23% exhibited symptoms of anxiety and/or depression, respectively (Figure 1A).

A logistic regression model, after controlling for age, BMI, and gender, revealed a significant association between asthma anxiety and depression. Specifically, individuals with asthma had an odds ratio of 2.40 (95% CI: 1.07–5.35) of having both anxiety and depression symptoms compared to non-asthmatics with no mood dysfunctions (Table 3). This association was not significant for individuals having either anxiety or depression symptoms. We also examined the association between asthma and anxiety or depression categories separately and found no significant association between asthma and anxiety after controlling for age, BMI, and gender (Appendix A). For depression, a significant relationship was revealed, where asthmatics had an odds ratio of 2.10 (95% CI: 1.00–4.43) of being in the borderline category compared to non-asthmatics categorized with no symptoms (Appendix A). A similar association was observed when combining both borderline and cases into one category, with a 1.99 (95% CI: 1.02–3.90) odds ratio for asthmatics having symptoms of depression (Appendix A). We performed a sensitivity analysis, adjusting for chronic health conditions (heart disease, hypertension, and diabetes), and the significant association between asthma and anxiety and depression symptoms (*p* = 0.04) persisted.

We also examined the association between asthma-perceived severity and anxiety and depression using logistic regression and found a significant association. Specifically, reported severe asthma increased the odds of having both anxiety and depression symptoms (OR = 10.83, 95% CI: 1.81–64.6; Appendix A).

### 3.3. Asthma Association with Cognition

The scores on the MoCA test were different between the two groups but did not reach statistical significance (F_(1,223)_ = 3.46, *p* = 0.06), where asthmatics had lower scores compared to non-asthmatics (Table 2). MCI was prevalent in 47% of this study population; the prevalence in asthmatics was 57% compared to non-asthmatics (42%; Figure 1B). Logistic regression analysis revealed a significant association between asthma and MCI after controlling for age, BMI, and gender, with an odds ratio of 1.80 (95% CI: 1.00–3.24; Table 3). We performed a sensitivity analysis, adjusting for chronic health conditions (heart disease, hypertension, and diabetes), and the significant association between asthma and MCI (*p* = 0.05) persisted.

### 3.4. Asthma Control Association with Anxiety

About 48% of asthmatics had been controlling their condition based on the ACT (Table 1). Only anxiety was significantly associated with asthma control, where the odds ratio of being categorized as a case was 4.88 (95% CI: 1.09–17.2) in poorly controlled asthma compared to well-controlled asthma (Table 4).

### 3.5. Participants’ Age Interval Moderating the Association between Asthma and Mood and Cognition

Middle-aged participants with asthma had a significant odds ratio of 2.5 (95% CI: 1.00–6.08) of having symptoms of both anxiety and depression compared to middle-aged non-asthmatics (Table 5). Asthmatic elderly participants were at significantly higher odds (7.4, 95% CI: 2.34–23.31) to have MCI compared to middle age non-asthmatics. Elderly non-asthmatic participants also had higher odds of having MCI but did not reach significance (Table 5).

## 4. Discussion

This study aimed to examine the effect of asthma on mental health including anxiety and depression and cognition in middle-aged and elderly individuals in Riyadh, Saudi Arabia. Our findings indicate a link between asthma and anxiety, depression, and MCI. We also observed that inadequate control of asthma was associated with increased levels of anxiety. The moderation of age on the relationship between anxiety, depression, and cognition varied among different groups. Specifically, middle-aged individuals with asthma displayed a higher likelihood of experiencing anxiety and depression. On the other hand, elderly subjects with asthma had an elevated risk of MCI.

A significant portion of our study’s asthmatic participants had adult-onset asthma. A previous study reported a remission rate of about 70% for asthma that manifests in childhood [34], providing a potential reason for the elevated incidence of adult-onset asthma in our analysis. Nevertheless, data is yet to be reported on the number of individuals who endure asthma from childhood into adulthood in Saudi Arabia.

Our study revealed that individuals with asthma had higher scores on the HAD scale, indicating higher levels of anxiety and depression compared to those without asthma. The prevalence of anxiety and depression among our study population was found to be 27% and 22%, respectively. These findings are in line with a similar study conducted in Riyadh, which investigated the correlation between anxiety, depression, and chronic illnesses in a primary healthcare setting [35]. While we did not observe a significant link solely between anxiety and asthma, our results indicated that individuals with asthma were twice as likely to experience depression compared to those without asthma. Another research study examining the association between asthma and anxiety and depression across 54 countries highlighted that patients diagnosed with asthma in the United Arab Emirates had higher rates of depression compared to those reporting anxiety [36]. It is well established that depression and anxiety frequently coexist as comorbid psychiatric conditions. The findings of this study also revealed that individuals with asthma were more than twice as likely to have both anxiety and depression. However, it is worth noting that some studies have indicated that the severity of asthma is associated with anxiety [37,38], which is a variable that was not objectively assessed in this particular study. Nonetheless, our findings also revealed that individuals who reported experiencing severe asthma were ten times more likely to exhibit symptoms of both anxiety and depression. It is worth noting that the perception of illness severity could potentially be influenced by coexisting conditions such as anxiety and depression. Therefore, in future research endeavors, it would be beneficial to explore this association further using objective measures of severity aligned with established clinical guidelines. The presence of anxiety and depression in asthmatic patients has been found to significantly impact their quality of life and utilization of healthcare resources. Research suggests that individuals with comorbid anxiety and depression experience a lower quality of life and are more likely to seek healthcare services related to their asthma [15]. This highlights the need to address the mental health aspects of asthma management and provide appropriate support to individuals with these comorbidities. By addressing the mental well-being of asthmatic patients, we can aim to reduce the burden of the disease and improve overall patient outcomes. Further studies are needed to explore effective interventions and strategies for managing asthma in individuals with mental health comorbidities.

We found that more than 50% of asthmatic individuals had uncontrolled asthma. Others have reported similar findings in Saudi Arabia at a range of 45–64% of uncontrolled asthma using the ACT [39,40]. An epidemiological study across the Middle East about asthma management reported that 70% of Saudi adults with asthma did not control their asthma [41]. Al Ghobain and colleagues also reported high rates of symptoms indicating uncontrolled asthma among adults in Saudi Arabia [4]. In our study, poorly controlled asthma was associated with higher odds of having anxiety but not depression. This association between anxiety and asthma control has been reported in several studies [37,38,42]. It is postulated that anxious asthmatic individuals might have an over perception of asthma symptoms due to their hypervigilance in fear of asthma attacks [43,44]. Furthermore, physical symptoms of anxiety overlap with asthma symptoms such as chest tightness and hyperventilation. The effects of anxiety and poor asthma control are observed at the structural level of the brain, where low grey matter volumes of the subcortical structures were reported [42]. This emphasizes the importance of introducing interventions targeting anxiety and improving asthma control. Improved asthma management was reported to reduce anxiety and improve asthma control [45]. Intervention studies using cognitive behavioral therapy and lifestyle changes show promising results in improving anxiety and asthma control [46,47]. Asthma control is associated with medication adherence [48], which is a measure that we did not examine in this study and could be a mediator in the relationship between anxiety and asthma control. Generally, studies indicate that there is no association between anxiety and adherence to medication in asthma [49,50], though there are other reports of lower adherence in patients of other chronic diseases with anxiety [51]. Others have highlighted that asthma-specific anxiety is more important in medication compliance [52]. Future work is needed to better understand the relationship between anxiety and drug adherence with asthma control.

In addition to anxiety and depression, MCI was observed at higher odds in asthmatic patients. The overall prevalence of MCI in our study population was 47%, which is comparable to a 45% prevalence in a study in Riyadh assessing the prevalence of cognitive impairment and dementia in adults over 60 years old [53]. Asthma was associated with MCI with an odds ratio of almost two. We did not find any association between asthma control and cognition. Other studies have also failed to find an association between MCI and asthma control [20,21]. Occasionally, elderly asthmatic patients may experience reduced perception of bronchial obstruction, a factor that potentially explains the lack of association with asthma control [54]. Furthermore, given that the ACT relies on self-reporting, there may be an inherent bias stemming from deficits in introspective capacities due to cognitive dysfunction. It is vital to address the association between asthma and early signs of MCI, as a recent report indicates that asthma augments the risk of dementia, especially in severe asthma [9]. Asthma also is associated with changes in the brain, including smaller hippocampal volume [22] and an increase in synaptic degeneration biomarkers [9]. One possible underlying condition is the inflammatory nature of the disease. Metrics of MRI scans of the brain of asthmatic patients indicated neuroinflammation [7]. Another study reported differences in hippocampal metabolites associated with reduced neuronal viability, glial activation, and reduced energy reserve [55].

This study reveals that the relationship between asthma and various aspects of mental health outcomes varied depending on the age group being studied. Specifically, middle-aged individuals with asthma were found to have 2.5 times higher odds of experiencing comorbid anxiety and depression, whereas there was no significant association observed with MCI. On the other hand, elderly patients with asthma had a seven-fold increase in their odds ratio for MCI, but no association was detected with anxiety and/or depression. The heightened likelihood of anxiety and depression during middle age could be attributed to the presence of life stressors at this stage, such as work demands and family responsibilities; it is worth noting that most participants in this study were employed and married. The majority of our participants were also women, who face hormonal changes that could put them at higher risk of anxiety and depression during this stage of life [56]. Moreover, there have been reports suggesting that with advancing age, individuals may exhibit a reduced susceptibility to the development of anxiety and depression. This could be potentially attributed to older adults having enhanced emotional control and decreased emotional reactivity [57]. As a result, it is not surprising to find the absence of an association between anxiety/depression symptoms in elderly patients. Notably, several studies indicate that a history of anxiety and/or depression can impact various cognitive functions such as memory decline, attention deficits, and impairment in executive functioning [58,59,60]. Consequently, we may speculate that asthmatic patients who experience anxiety or depression during middle age years are more susceptible to developing cognitive impairments as they progress into older age. Future research is needed to explore this hypothesis.

There are some limitations of this study. One significant drawback of our study is the relatively small sample size, which hampers the generalizability of our findings to the broader population in Riyadh and Saudi Arabia. This limitation could be attributed to the fact that our study was conducted at a single center and utilized convenience sampling. The composition of our participant cohort largely consisted of educated, married women, which was as expected, given that we recruited from King Abdullah University Hospital associated with Princess Nourah bint Abdulrahman University—a female-only institution catering primarily to university staff, students, and their families. It should be noted that these participant characteristics pose another limitation when generalizing the findings from this study. Additionally, restricting our sample pool to adults who are middle-aged and elderly further constrained the sample size. Nevertheless, our findings are in line with other research findings, which provides legitimacy to our study. Additionally, while our focus was on a specific sub-population of adults (middle-aged and elderly adults), resulting in a smaller participant pool, it allowed for greater specificity and precision in our analysis. We also implemented appropriate statistical controls for covariates and stratified the data by age groups, revealing statistically significant associations that reflect the rigor of our study. Nonetheless, future studies should aim to bolster sample sizes through multi-center recruitment efforts building on the findings in this study to ensure the replicability of these findings and draw more confident conclusions of the reported associations found here. Secondly, we did not collect any clinical characteristics of asthmatic patients, such as lung functioning and asthma symptoms, that could help better understand the relationships investigated. The potential impact of comorbidities associated with other chronic health conditions on mental health outcomes was not fully explored in this study. Future research will investigate the influence of multiple comorbidities on mental health outcomes. Nevertheless, this study is the first to highlight the impact of asthma on mental well-being in middle-aged and elderly individuals in Saudi Arabia. Our results align with previous studies conducted in other countries, further illuminating the impact of asthma on various dimensions of mental well-being throughout different age groups. These findings emphasize the need for regular screening to assess psychological health among individuals with asthma and highlight the importance of implementing targeted interventions to promote overall well-being.

## Figures and Tables

**Figure 1 behavsci-13-00842-f001:**
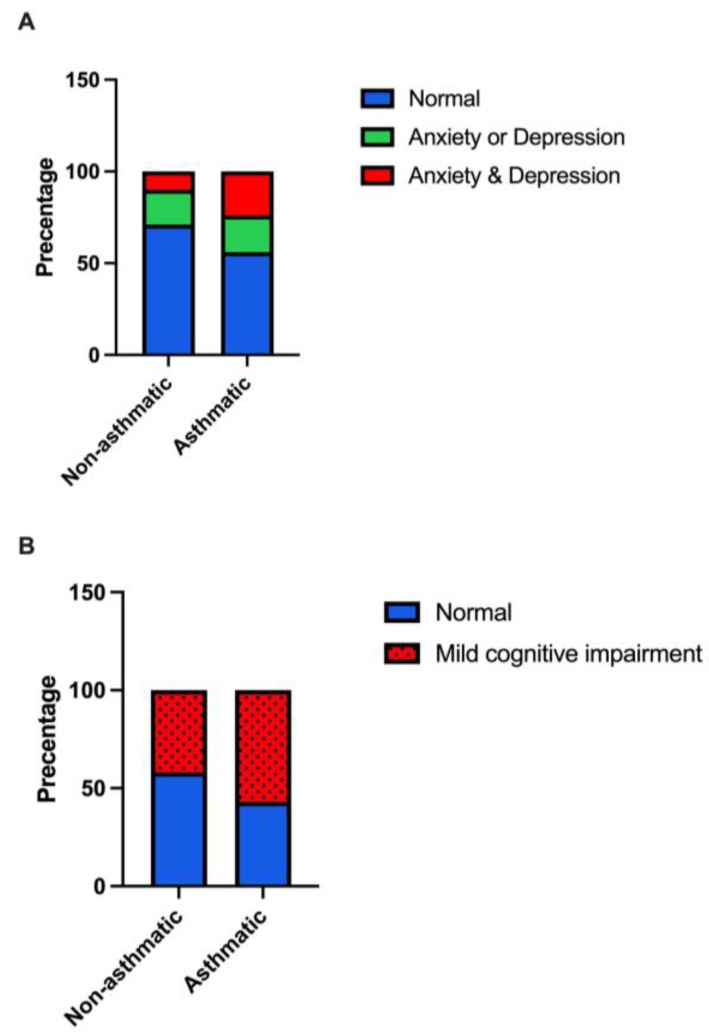
The prevalence of (**A**) anxiety and/or depression, and (**B**) mild cognitive impairment among asthmatic compared to non-asthmatic individuals.

**Table 1 behavsci-13-00842-t001:** Study participants’ characteristics.

Variables		Mean (Range)	Frequency	Percent (%)
Age		52.3 (40–81)		
	Middle age (40–59)		148	75.7
	Elderly (≥60)		59	24.3
BMI		31 (16.6–50.9)		
Sex				
	Male		55	22.6
	Female		188	77.4
Social status				
	Single		8	3.3
	Married		211	86.8
	Divorced		15	6.2
	Widow		9	3.7
Education				
	None		21	8.6
	Elementary		19	7.8
	Middle		15	6.2
	Secondary		40	16.5
	University		92	37.9
	Postgraduate		56	23.0
Occupation				
	Unemployed		60	24.7
	Employed		146	60.1
	Retired		37	15.2
Asthmatic				
	Yes		84	34.6
	No		159	65.4
Onset of Asthma				
	<12 years old		10	11.9
	12–18 years old		10	11.9
	>18 years old		64	76.2
Asthma Severity				
	Mild		25	29.8
	Moderate		43	51.2
	Sever		16	19.0
Asthma Control				
	Well-controlled		40	47.6
	Partially controlled		15	17.9
	Poorly controlled		29	34.5
Chronic health conditions				
Heart disease			17	7
Hypertension			91	37
Diabetes			78	32

**Table 2 behavsci-13-00842-t002:** Mean scores (±SD) of anxiety, depression, and cognition using study questionnaires and its associations with asthma.

Variables	Asthmatic (n = 84)	Non-Asthmatic (n = 159)	Total (n = 243)	*p*-Value ^b^
Anxiety	6.2 ± 4.7	4.7 ± 3.5	5.2 ± 4.0	0.04
Depression	5.2± 3.9	4.1 ± 3.3	4.5 ± 3.6	0.08
MoCA Test ^a^	23.7 ± 5.0	25.5 ± 3.7	24.9 ± 4.2	0.06

^a^ Four participants did not complete the MoCA test, ^b^ Univariate ANOVA controlling for age, gender, and BMI as covariates.

**Table 3 behavsci-13-00842-t003:** Multinomial logistic regression on the association between asthma and anxiety and/or depression and cognition, while controlling for covariates.

Variables	Anxiety/Depression				Cognition	
	Anxiety or Depression		Anxiety and Depression		Mild Cognitive Impairment	
	OR (CI 95%)	*p*-Value	OR (CI 95%)	*p*-Value	OR (CI 95%)	*p*-Value
Age	0.99 (0.95–1.03)	0.8	0.98 (0.94–1.02)	0.3	1.03 (1.00–1.06)	0.06
Gender (Female)	1.33 (0.57–3.10)	0.5	1.80 (0.57- 5.70)	0.3	0.80 (0.41–1.56)	0.5
BMI	0.99 (0.93–1.06)	0.8	1.06 (0.99–1.13)	0.08	1.03 (0.98–1.08)	0.2
Asthma	1.33 (0.62–2.77)	0.4	**2.40** **(1.07–5.35)**	**0.03**	**1.80** **(1.00–3.24)**	**0.05**

**Table 4 behavsci-13-00842-t004:** Multinomial logistic regression on the association between asthma control and anxiety.

Variables	Anxiety			
	Borderline		Case	
	OR (CI 95%)	*p*-Value	OR (CI 95%)	*p*-Value
Age	0.95 (0.88–1.02)	0.1	0.96 (0.90–1.03)	0.2
Gender (Female)	1.20 (0.21–6.82)	0.8	1.86 (0.19–18.3)	0.6
BMI	0.98 (0.89–1.09)	0.7	1.05 (0.96–1.16)	0.3
Asthma control (Well controlled)				
Poorly controlled	0.92 (0.19–4.33)	0.9	**4.33** **(1.09–17.2)**	**0.04**
Not well controlled	1.46 (0.30–7.10)	0.6	1.89 (0.28–12.5)	0.5

**Table 5 behavsci-13-00842-t005:** Multinomial logistic regression on the association between age and asthma and anxiety, depression, and cognition.

Variables	Anxiety/Depression				Cognition	
	Anxiety or Depression		Anxiety and Depression		Mild Cognitive Impairment	
	OR (CI 95%)	*p*-Value	OR (CI 95%)	*p*-Value	OR (CI 95%)	*p*-Value
Gender (Female)	1.33 (0.57–3.08)	0.5	1.80 (0.57–5.70)	0.3	0.80 (0.41–1.56)	0.5
BMI	0.99 (0.93–1.05)	0.8	1.06 (0.99–1.13)	0.08	1.03 (0.98–1.08)	0.2
Asthma age interval (Middle age Non-asthmatic)						
Middle age asthmatic	1.83 (0.80–4.18)	0.1	**2.48** **(1.00–6.08)**	**0.05**	1.44 (0.74–2.81)	0.3
Elderly non-asthmatic	1.61 (0.59–4.37)	0.3	0.73 (0.15–3.53)	0.7	2.07 (0.88–4.82)	0.09
Elderly asthmatic	0.77 (0.20–2.96)	0.7	1.60 (0.48–5.36)	0.4	**7.39** **(2.34–23.31)**	**0.001**

## Data Availability

The data used in this study are available and will be provided by the corresponding author at reasonable request.

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
