# Peer review of "The Association of Asthma with Anxiety, Depression, and Mild Cognitive Impairment among Middle-Aged and Elderly Individuals in Saudi Arabia"

_behavsci, 2023, doi:10.3390/bs13100842_

Round 1

Reviewer 1 Report

- The abstract should be written in more academic way. Odds ratio with 95% confidence interval should be written. More methodological details should be written in the abstract including the settings, study time frame, and main statistical analysis used.

- First heading in the method section "participants" is not appropriate to be used as it is doesn't cover all information presented afterward. The method section should have the following main subheadings: study design, study population and settings, participants recruitment, study instruments, piloting and validity check, translation, ethical approval, and sample size calculation. Then, detailed information should be presented for each subheading.

- The statistical analysis should be re-checked. The choice of using parametric test depends on the kurtosis, skewness, and normality of the histogram. This is critical in determining the central tendency measure to be used while presenting the depression and anxiety score. Then, it will affect the implementation of regression analysis.

- Building the whole study on 84 asthmatic patients only is not right. I am sure that the study findings are driven by non-asthamtic patients, which is a major limitation flow in the methodology.

Needs revision

Author Response

We thank the reviewers for their comments, which we address in our point-by-point response. We believe the manuscript has been greatly strengthened as a result. 

  1. The abstract should be written in more academic way. Odds ratio with 95% confidence interval should be written. More methodological details should be written in the abstract including the settings, study time frame, and main statistical analysis used.
  • The results were rewritten to include the odds ratios along with the confidence intervals in the abstract. In addition, we have clarified that logistic regression was used as the main statistical analysis while controlling for covariates. In order to keep with the word count limit of the journal of 200 words in the abstract, we included only that patients were recruited from outpatient clinics to indicate the study settings.
  1. First heading in the method section "participants" is not appropriate to be used as it is doesn't cover all information presented afterward. The method section should have the following main subheadings: study design, study population and settings, participants recruitment, study instruments, piloting and validity check, translation, ethical approval, and sample size calculation. Then, detailed information should be presented for each subheading.
  • The methods section has been modified to include study design and setting, study population, study procedure, and study instruments as recommended.
  1. The statistical analysis should be re-checked. The choice of using parametric test depends on the kurtosis, skewness, and normality of the histogram. This is critical in determining the central tendency measure to be used while presenting the depression and anxiety score. Then, it will affect the implementation of regression analysis.
  • Data distribution did not show substantial departure from normality when examining histograms and Q-Q plots. In addition, skewness and kurtosis values were all below one, indicating no extreme departure from normality [1]. Moreover, with sample sizes > 30, studies suggest that parametric tests would not be greatly impacted with moderate deviation from normality [2]. This is now added to the methods section.

This i now added to the statistical analysis section of the methods

  1. Building the whole study on 84 asthmatic patients only is not right. I am sure that the study findings are driven by non-asthamtic patients, which is a major limitation flow in the methodology.
  • We acknowledge the limitation of our sample size. This unequal sample sizes among control and cases groups is an issue in observational studies. We have conducted an a priori sample size calculation was done using the G*Power sample size calculator with a 5% type one error, 95% confidence level, and 80% study power. The calculated sample size required for this study was 194, with 97 participants in each asthmatic and control group. Assuming a higher incidence of controls, a larger sample size of 245 was considered. To ensure a good representation of the sample, a power analysis was conducted with the final sample size obtained and it indicated an 85% power depicting good representativeness of the sample. Inclusion criteria were men and women 40 years of age and older with doctor-diagnosed asthma for asthmatic participants. Patients suffering from a severe chronic illness requiring bed rest, physical disability, and the presence of communication barriers were excluded.

This is now added to the population section of the methods.

References:

  1. Kim, H.-Y. Statistical notes for clinical researchers: assessing normal distribution using skewness and kurtosis. Restor. Dent. Endod. 2013, 38, 52, doi:10.5395/rde.2013.38.1.52.
  2. Ghasemi, A.; Zahediasl, S. Normality tests for statistical analysis: A guide for non-statisticians. Int. J. Endocrinol. Metab. 2012, 10, 486–489, doi:10.5812/ijem.3505.

Reviewer 2 Report

The Authors analyzed, in a group of asthmatic patients, the possible association with anxiety, depression, and cognitive impairment among middle-aged and elderly individuals in Saudi Arabia. The data are already known in previous publications, they are original in relation to the population considered.

Some clarifications:

- anxiety and other parameters are related to poor asthma control. Is there a correlation also with the classification of asthma severity?

- Is it possible that anxiety affects adherence, resulting in poor asthma control?

- what is the possible explanation of the mild cognitive impairment?

Author Response

We thank the reviewers for their comments, which we address in our point-by-point response. We believe the manuscript has been greatly strengthened as a result. 

Below are the reviewers’ comments along the responses. For clarity the reviewers’ comments are in bold and our responses are in regular font.

Reviewer 2:

The Authors analyzed, in a group of asthmatic patients, the possible association with anxiety, depression, and cognitive impairment among middle-aged and elderly individuals in Saudi Arabia. The data are already known in previous publications, they are original in relation to the population considered.

Some clarifications:

  1. anxiety and other parameters are related to poor asthma control. Is there a correlation also with the classification of asthma severity?
  • Unfortunately, we did not have an objective measure of asthma severity in this study using recommended guidelines by the Saudi Initiative for Asthma. However, our questionnaire included a question about patients’ perception of asthma where they could answer as mild, moderate, or sever. We examined the association between asthma perceived severity and anxiety and depression using logistic regression and found a significant association. Specifically, reported severe asthma increased the odds of having both anxiety and depression symptom (OR= 10.83, 95% CI: 1.81-64.6; Supp. Table 3). No association were seen with cognition.
  • Our findings also revealed that individuals who reported experiencing severe asthma were ten times more likely to exhibit symptoms of both anxiety and depression. It is worth noting that the perception of illness severity could potentially be influenced by coexisting conditions such as anxiety and depression. Therefore, in future research endeavors, it would be beneficial to explore this association further using objective measures of severity aligned with established clinical guidelines.

This is now is added to the results and discussion sections. These findings are extremely valuable and future studies will examine this association further using more objective measure of severity following recommended gridlines. We thank the reviewer for their insights and recommendation.   

  1. Is it possible that anxiety affects adherence, resulting in poor asthma control?
  • Asthma control is associated with medication adherence [1], which is a measure that we did not collect in this study and could be a mediator in the relationship between anxiety and asthma control. Generally, studies indicate that there is no association between anxiety and adherence to medication in asthma [2,3], though there are other reports of lower adherence in patients of other chronic diseases with anxiety [4]. Others have highlighted that asthma-specific anxiety is more important in medication compliance [5]. Future work is needed to better understand the relationship between anxiety and drug adherence with asthma control.

This is now added to the discussion section.

  1. what is the possible explanation of the mild cognitive impairment?
  • One possible underlying condition is the inflammatory nature of the disease. Metrics of MRI scans of the brain of asthmatic patients indicated neuroinflammation [6]. Another study reported differences in hippocampal metabolites associated with reduced neuronal viability, glial activation, and reduced energy reserve [7].

This is now added to the discussion section.

References:

  1. Engelkes, M.; Janssens, H.M.; de Jongste, J.C.; Sturkenboom, M.C.J.M.; Verhamme, K.M.C. Medication adherence and the risk of severe asthma exacerbations: a systematic review. Eur. Respir. J. 2015, 45, 396–407, doi:10.1183/09031936.00075614.
  2. Toelle, B.G.; Marks, G.B.; Dunn, S.M. Psychological and Medical Characteristics Associated with Non-Adherence to Prescribed Daily Inhaled Corticosteroid. J. Pers. Med. 2020, 10, 126, doi:10.3390/jpm10030126.
  3. Bosley, C.; Fosbury, J.; Cochrane, G. The psychological factors associated with poor compliance with treatment in asthma. Eur. Respir. J. 1995, 8, 899–904, doi:10.1183/09031936.95.08060899.
  4. DiMatteo, M.R.; Lepper, H.S.; Croghan, T.W. Depression Is a Risk Factor for Noncompliance With Medical Treatment. Arch. Intern. Med. 2000, 160, 2101, doi:10.1001/archinte.160.14.2101.
  5. Thoren, C.; Petermann, F. Reviewing asthma and anxiety. Respir. Med. 2000, 94, 409–415, doi:10.1053/rmed.1999.0757.
  6. Rosenkranz, M.A.; Evans, M.D.; Mumford, J.A.; Esnault, S.; Davidson, R.J.; Busse, W.W. The Effect of Asthma on Activation of Brain Neurocircuits. J. Allergy Clin. Immunol. 2019, 143, AB7, doi:10.1016/j.jaci.2018.12.021.
  7. Kroll, J.L.; Steele, A.M.; Pinkham, A.E.; Choi, C.; Khan, D.A.; Patel, S. V.; Chen, J.R.; Aslan, S.; Sherwood Brown, E.; Ritz, T. Hippocampal metabolites in asthma and their implications for cognitive function. NeuroImage Clin. 2018, 19, 213–221, doi:10.1016/j.nicl.2018.04.012.

Reviewer 3 Report

This is a cross-sectional study on the association between asthma and mental health (anxiety, depression and cognitive impairment) in Saudi Arabia, in which the authors have found that asthmatics had 2 and 1.8 times the odds of having both anxiety and depression and mild cognitive impairment, respectively, with more pronounced effects in poorly controlled asthma, confirming recent research results for the study population. 

The paper is well written, the sample size and statistical methods are adequate. While the authors have acknowledged the lack of clinical characteristics as a limitation of their study, I would recommend to include in the discussion that this doens’t just entail lung function, but also general information on comorbidities (e. g. Charlson comorbidity index). While adjusting for age might include by proxy a certain burden of comorbidity, it would be interesting to see whether the association between asthma and cognitive impairment persists when adjusting for vascular diseases and risk factors, such coronary heart disease, diabetes and smoking. It should also be noted that normality should not only be assumed based on the sample size, but also on the distribution of target variables (line 114).

Author Response

We thank the reviewers for their comments, which we address in our point-by-point response. We believe the manuscript has been greatly strengthened as a result. 

Below are the reviewers’ comments along the responses. For clarity the reviewers’ comments are in bold and our responses are in regular font.

Reviewer 3:

This is a cross-sectional study on the association between asthma and mental health (anxiety, depression and cognitive impairment) in Saudi Arabia, in which the authors have found that asthmatics had 2 and 1.8 times the odds of having both anxiety and depression and mild cognitive impairment, respectively, with more pronounced effects in poorly controlled asthma, confirming recent research results for the study population. 

The paper is well written, the sample size and statistical methods are adequate.

  1. While the authors have acknowledged the lack of clinical characteristics as a limitation of their study, I would recommend to include in the discussion that this doens’t just entail lung function, but also general information on comorbidities (e. g. Charlson comorbidity index).
  • The potential impact of comorbidities associated with other chronic health conditions on mental health outcomes was not fully explored in this study. Future research will investigate the influence of multiple comorbidities on mental health outcomes.

This is now added to the study limitation paragraph in the discussion section.

  1. While adjusting for age might include by proxy a certain burden of comorbidity, it would be interesting to see whether the association between asthma and cognitive impairment persists when adjusting for vascular diseases and risk factors, such coronary heart disease, diabetes and smoking.
  • We have performed a sensitivity analysis adjusting for heart disease, hypertension, and diabetes and the significant association between asthma and anxiety and depression symptoms (P= 0.04) and mild cognitive impairment (P= 0.05) persisted.

This sentence was added the results section. We have also included the prevalence of these chronic condition in Table 1.  

  1. It should also be noted that normality should not only be assumed based on the sample size, but also on the distribution of target variables (line 114).
  • Data distribution did not show substantial departure from normality when examining histograms and Q-Q plots. In addition, skewness and kurtosis values were all below one, indicating no extreme departure from normality [1]. Moreover, with sample sizes > 30, studies suggest that parametric tests would not be greatly impacted with moderate deviation from normality [2]. This is now added to the methods section.

This now added to the statistical analysis section of the methods.

Reference:

  1. Kim, H.-Y. Statistical notes for clinical researchers: assessing normal distribution using skewness and kurtosis. Restor. Dent. Endod. 2013, 38, 52, doi:10.5395/rde.2013.38.1.52.
  2. Ghasemi, A.; Zahediasl, S. Normality tests for statistical analysis: A guide for non-statisticians. Int. J. Endocrinol. Metab. 2012, 10, 486–489, doi:10.5812/ijem.3505.

Reviewer 4 Report

The study examined associations between asthma and several mental health indicators among Saudi Arabia population. I think the paper has potential to make a contribution to the literature given that some revisions are made.

1.       Throughout the paper, the authors used the term “control” in two different meanings, which may lead to misunderstanding. Among participants, there are asthma patients and control groups; while among asthma patients, there is a degree of control (i.e., use of medicine). Using terminology that is clearer may avoid potential confusion.

2.       No hypothesis was raised about the role of asthma control in how asthma may lead to anxiety or depression.

3.       In the control group of 159 participants, what are the criteria for their inclusion? Since the main findings regarding asthma patients’ mental health was based on comparison with this control group, their representativeness is important. Moreover, the authors should report the sample size determination process, e.g., power analysis.

4.       Line 149-153. The way ANOVA results were reported is a bit strange. ANOVA was used to determine group differences, but here the results were reported as if they could somehow determine “association”.

5.       In the univariate ANOVAs, the authors controlled age, gender and BMI. It would be better if they could provide a version without these controls and to see if these controls changed the results. Also, including these three variables as control should be based on evidence.

6.       Apart from ANOVA, the authors proceeded to conduct a series of logistic regression. The choice of these analysis methods warrants a brief explanation.

7.       In Discussion section, the second paragraph (Lines 243 - 252) seems misplaced. It should be moved to limitation or other places. 

Author Response

We thank the reviewers for their comments, which we address in our point-by-point response. We believe the manuscript has been greatly strengthened as a result. 

Below are the reviewers’ comments along the responses. For clarity the reviewers’ comments are in bold and our responses are in regular font.

Reviewer 4:

  1. Throughout the paper, the authors used the term “control” in two different meanings, which may lead to misunderstanding. Among participants, there are asthma patients and control groups; while among asthma patients, there is a degree of control (i.e., use of medicine). Using terminology that is clearer may avoid potential confusion.
  • We have changed the term “control” study participants to non-asthmatic throughout the manuscript. Therefore, now the term “control” is only used to describe the degree of control of asthma in asthmatic participants.
  1. No hypothesis was raised about the role of asthma control in how asthma may lead to anxiety or depression.
  • We also hypothesized that poor asthma control would be associated with worse mental health outcomes.

This sentence was added to the introduction.

  1. In the control group of 159 participants, what are the criteria for their inclusion? Since the main findings regarding asthma patients’ mental health was based on comparison with this control group, their representativeness is important. Moreover, the authors should report the sample size determination process, e.g., power analysis.
  • We acknowledge the limitation of our sample size. This unequal sample sizes among control and cases groups is an issue in observational studies. We have conducted an a priori sample size calculation was done using the G*Power sample size calculator with a 5% type one error, 95% confidence level, and 80% study power. The calculated sample size required for this study was 194, with 97 participants in each asthmatic and control group. Assuming a higher incidence of controls, a larger sample size of 245 was considered. To ensure a good representation of the sample, a power analysis was conducted with the final sample size obtained and it indicated an 85% power depicting good representativeness of the sample. Inclusion criteria were men and women 40 years of age and older with doctor-diagnosed asthma for asthmatic participants. Patients suffering from a severe chronic illness requiring bed rest, physical disability, and the presence of communication barriers were excluded.

This now added to the population section of the methods.

  1. Line 149-153. The way ANOVA results were reported is a bit strange. ANOVA was used to determine group differences, but here the results were reported as if they could somehow determine “association”.
  • The reported results were modified to indicate only group differences and not association accordingly.
  1. In the univariate ANOVAs, the authors controlled age, gender and BMI. It would be better if they could provide a version without these controls and to see if these controls changed the results. Also, including these three variables as control should be based on evidence.
  • To examine the potential relationship between age and BMI with anxiety, depression, and MCI, a preliminary correlation analysis was conducted, revealing significant correlations. Therefore, age and BMI were included as covariates in all regression models. Additionally, there exists extensive literature demonstrating a link between gender and psychological well-being—with women being disproportionately affected by anxiety, depression, and cognitive impairment [1,2]. As a result of this evidence, gender was also included as a covariate in all regression models.
  • Below is the table with the ANOVA results without controlling for the covariates

Variables

Asthmatic

(n=84)

Non-asthmatic

(n=159)

Total (n=243)

P-value b

Anxiety

6.2 ± 4.7

4.7 ± 3.5

5.2 ± 4.0

0.004

Depression

5.2± 3.9

4.1 ± 3.3

4.5 ± 3.6

0.01

MoCA Testa

23.7 ± 5.0

25.5 ± 3.7

24.9 ± 4.2

0.004

The differences are highly significant which is expected without controlling for the covariates.

This table is added to the supplementary material.

  1. Apart from ANOVA, the authors proceeded to conduct a series of logistic regression. The choice of these analysis methods warrants a brief explanation.
  • A brief explanation is added below and highlighted

A multinomial logistic regression was carried out to assess the association between asthma and anxiety and depression categories with the normal category as the reference. Another model tested the association between asthma and the presence of any symptoms of anxiety or depression by combining both categories, borderline, and cases, into one category, which consider any deviation from normal scores and to increase power combining the two abnormal groups. In addition, we tested the association between asthma and having either anxiety or depression or both using multinomial regression to address the common comorbidity of both psychological disorders.

  1. In Discussion section, the second paragraph (Lines 243 - 252) seems misplaced. It should be moved to limitation or other places.

These sentences are explaining and discussing the descriptive statistics of the demographics of the study that were presented at the beginning of the study, therefore they were presented first in the discussion as well.

References:

  1. Pavlidi, P.; Kokras, N.; Dalla, C. Sex Differences in Depression and Anxiety. In; 2022; pp. 103–132.
  2. Lin, K.A.; Choudhury, K.R.; Rathakrishnan, B.G.; Marks, D.M.; Petrella, J.R.; Doraiswamy, P.M. Marked gender differences in progression of mild cognitive impairment over 8 years. Alzheimer’s Dement. Transl. Res. Clin. Interv.2015, 1, 103–110, doi:10.1016/j.trci.2015.07.001.

Round 2

Reviewer 1 Report

The study sample is not sufficient to draw this conclusion 

Author Response

We appreciate the feedback and revisions provided by the editor and reviewers, which have significantly strengthened our manuscript. We have diligently addressed all comments from the reviewers. To address this concern, we provided evidence of a priori sample size calculations and power analyses that support our current sample sizes as sufficient for drawing conclusions in this study. Our findings align with previous research, lending credibility to our study. While our focus was on a specific sub-population (middle-aged and elderly adults), resulting in a smaller participant pool, it allowed for greater specificity and precision in our analysis. We also implemented appropriate statistical controls for covariates and stratified the data by age groups revealing statistically significant associations that reflect the rigor of our study design. Looking ahead, future studies will aim to bolster sample sizes through multi-center recruitment efforts.

Reviewer 4 Report

The authors have addressed the main concerns. I have no further comments.

Author Response

We thank you and the reviewers for the thorough revisions of this manuscript which we believe has strengthened our paper as a result.